# Arm weight effects on dynamic walking stability in individuals with hemiparetic stroke

Hyung Suk Yang[ID][1]*, Lee T. Atkins[2], C. Roger James[2]

1 Division of Kinesiology and Sport Management, University of South Dakota, Vermillion, SD, United States of America, 2 Center for Rehabilitation Research, Texas Tech University Health Sciences Center, Lubbock, TX, United States of America

* HS.Yang@usd.edu

**Data Availability Statement:** All relevant data are within the manuscript and its Supporting Information files.

## Abstract

This study examined the effects of arm weights on dynamic stability during overground walking in individuals with hemiparetic stroke. Arm weights have been shown to improve mobility in stroke survivors, potentially at the cost of decreased dynamic stability and increased fall risk. Data from nine stroke survivors (8 males, 1 female; age: 58.0 ± 6.8 years) were assessed under four conditions: no weight, weight attached to the non-hemiparetic side, weight attached to the hemiparetic side, and bilateral weights. Each condition used 0.45 kg sandbags. Kinematic data were captured using an eight-camera motion system and analyzed to assess center of mass position and velocity relative to the base of support. Although repeated measures ANOVA showed no significant differences in stability across conditions, individual scatter plots revealed variable responses among participants. Some maintained or improved their stability, while others experienced decreases under specific conditions. These findings underscore the need for personalized approaches in rehabilitation planning, suggesting that integrating arm weights into rehabilitation protocols may not compromise dynamic stability for most stroke survivors. Further research with larger sample sizes and varied weights is essential to validate these findings and tailor the use of arm weights in stroke rehabilitation more effectively.

## Introduction

Stroke is a leading cause of long-term disability worldwide, with hemiparetic stroke significantly impairing motor control and walking abilities [1]. Consequently, to optimize rehabilitation outcomes, one must achieve sufficient dynamic walking stability to reduce their risk of falling, and further hindering recovery. Among various methods to assess walking stability, one approach involves examining the position and velocity of the body's center of mass (COM) in relation to the base of support (BOS) [2, 3]. For a given COM position, a backward loss of balance will occur if the forward velocity relative to the BOS is below the computational threshold for maintaining balance. Conversely, a forward loss of balance will occur if the COM forward velocity relative to the BOS is above the threshold.

Current rehabilitation strategies for stroke patients often emphasize improving strength, coordination, and gait mechanics [4]. Gait training is a central component, utilizing methods

**Funding:** The author(s) received no specific funding for this work.

**Competing interests:** The authors have declared that no competing interests exist.

such as treadmill training combined with body-weight support systems to enable patients to practice walking in a controlled environment [5]. Speed-dependent treadmill training further enhances walking performance and adaptability, with variations in treadmill speed helping to improve gait mechanics and confidence [6, 7]. Balance training is also essential, addressing both static and dynamic balance through exercises such as standing on one leg or navigating obstacle courses to improve stability during movement [8, 9].

Neuromuscular re-education techniques, including functional electrical stimulation and Constraint-Induced Movement Therapy, facilitate motor recovery by activating muscles and promoting the use of affected limbs [10, 11]. Advances in rehabilitation technology, such as virtual reality systems and robotic devices, offer innovative ways to practice movements and enhance motor skills. Virtual reality provides immersive environments for interactive training, while robotic-assisted gait training devices support walking practice [12, 13].

Sensory re-education is another intervention strategy for stroke survivors that focuses on proprioceptive training to enhance body position awareness and visual-vestibular training to improve balance [14, 15]. Despite these comprehensive approaches, limitations persist in fully addressing the complexities of dynamic walking stability changes that could affect fall risk. Recent studies suggest that applying lighter weights (e.g., 0.45 kg) to the arms can enhance walking performance in both healthy individuals and those affected by stroke [16, 17]. Reported enhancements include improved walking speed, step length, and cadence, likely due to enhanced proprioceptive feedback, passive mechanical energy transfer, and cognitive influences.

Despite these reported enhancements, the impact of arm weights on walking stability, particularly among individuals with hemiparetic stroke, remains unknown. It is crucial for clinicians and researchers to understand if, and to what extent, the addition of arm weights influence dynamic walking stability. Additionally, this information can aid in the development of optimal interventions to reduce fall risk and promote effective recovery. Although arm weights previously improved walking speed [16], their use as a rehabilitation intervention in stroke survivors could be contraindicated if dynamic stability decreases. If stability is not negatively affected, the addition of arm weights could be an effective, safe, low-cost, and easy-to-implement conservative intervention. By exploring the effects of arm weights on dynamic stability, this study aims to provide insights into potential therapeutic approaches that could enhance walking stability and overall mobility for stroke survivors. Therefore, the purpose of this study was to explore the effects of arm weights on dynamic stability among individuals affected by hemiparetic stroke. We hypothesized that because the addition of 0.45 kg arm weights is relatively light and only a small percentage of body weight, it would not alter the body's dynamic COM and BOS relationship, and thus not decrease dynamic walking stability. This Phase I trial [18] was focused on safety and further establishing proof-of-concept and feasibility of a potential mobility-enhancing treatment to improve activity participation and quality of life in stroke survivors.

## Methods

### Participants

This study analyzed gait performance data previously collected by Yang and colleagues (2023) from a group of individuals with stroke. Detailed protocol of the data collection for this study can be found elsewhere [16]. The study involved nine individuals with hemiparetic stroke (8 males, 1 female; age: 58.0 ± 6.8 years, height: 1.74 ± 0.07 m, mass: 78.9 ± 12.0 kg, BMI: 25.9 ± 3.8, stroke duration: 91.4 ± 67.9 months). These participants were volunteers with a history of hemiparetic stroke, aged between 40 and 70 years. Inclusion criteria required

participants to walk without a gait aid for at least 6 meters consecutively across multiple trials and to tolerate the addition of 0.45 kg weights on both arms. Exclusion criteria included a history of shoulder subluxation, significant joint pathology in the upper or lower limbs (such as peripheral neurological, rheumatic, orthopedic, or cardiovascular conditions that could interfere with gait), inability to follow verbal instructions, recent Botulinum toxin injections in the three months prior to testing, obesity (BMI >30 kg/m$^2$), and self-reported pregnancy. Additionally, participants were excluded if they could walk faster than 1.0 m/s or exhibited normal bilateral step length, swing time, or stance time symmetry [16]. Somatosensory impairments were also measured to exclude participants with peripheral neuropathy, as such impairments may further decrease gait performance, which is beyond the scope of this study [19]. The somatosensory screening used a Weinstein Enhanced Sensory Test monofilament (5.07-gauge wire; Semmes Weinstein Corp., Riverdale, NY) with 10 g of force. The measurement sites were the palmar surface of the index finger and thumb, the little finger and hypothenar eminence, the hand's dorsal surface between the second and third metacarpal heads, the first toe's plantar surface, the fifth toe's plantar surface, the dorsal surface between the first and second metatarsal heads, the third metatarsal head, and the fifth metatarsal base [20]. If a participant could not feel the touch in any location, indicating a loss of approximately 98% of sensory ability [21], they would have been excluded from the study. All participants met the inclusion criteria, and none were excluded. The study was approved by the Institutional Review Board of the Texas Tech University Health Sciences Center (IRB number: L16-131; date of approval: June 1, 2016). All participants provided written informed consent prior to participation. Participants were recruited in the period from 06.01.16 to 04.23.18.

## Protocol

Participants walked over-ground at their preferred speed during four different conditions: no weight (C1), non-hemiparetic side weight (C2), hemiparetic side weight (C3), and bilateral weights (C4). In the weighted conditions, participants carried a 0.45 kg sandbag on one or both wrists. The sandbag weight was placed on the anterior/palmar side of the wrist, over the distal one-third of the forearm, and secured with SuperWrap (Fabrifoam, Exton, PA). A 0.45 kg sandbag weight was chosen based on studies indicating that lighter weights improve walking performance through enhanced proprioceptive feedback and passive mechanical energy transfer [16, 17]. The weight was selected to provide a manageable perturbation while ensuring participant comfort. Participants were instructed to "walk normally at your usual speed" to initiate gait, without any additional instructions or feedback from the investigators. To minimize the potential confounding effects of learning, the order of testing conditions was randomized, and participants were allowed to perform several practice trials. Additionally, participants wore standardized lab attire, including athletic shorts, shirt, and shoes.

## Instrumentation and data analysis

Kinematics were captured (100 Hz) in three dimensions using an eight-camera motion capture system (Vicon, 2.2.3, Denver, CO). Thirty-nine 14 mm reflective markers were positioned according to Vicon's full Body Plug-in-Gait marker set.

Kinematic data were exported from Vicon Nexus to Matlab (The Mathworks, Inc., Natick, MA) for further processing. The three-dimensional marker coordinates were filtered using a fourth-order, zero-lag Butterworth low-pass digital filter with a cutoff frequency of 6 Hz. Pelvis markers were utilized to compute the pelvis center, estimating the whole-body COM [22]. Subsequently, the position and velocity of the estimated COM (i.e., COM state) were calculated relative to the BOS in the walking direction, which was determined by the location of the

heel marker and foot length. Foot length was estimated based on body height [23]. The relative position and velocity of the COM were normalized to foot length and the square root of gravitational acceleration multiplied by body height, respectively [3]. Normalized COM velocity was graphed against its position at each gait event and quantified relative to the simulation generated Feasible Stability Region (FSR) [2, 3]. The FSR method was chosen because, compared to other stability measures like Floquet multipliers, Lyapunov exponents, and margin of stability, FSR is more sensitive to fall risk as it directly evaluates the critical relationship between the COM and BOS, making it especially useful for assessing dynamic stability in complex tasks [24]. The gait events included initial contact and end-of-contact bilaterally, and were detected using an algorithm that included the sagittal plane velocity of the heel and metatarsal markers as input variables [25, 26]. Stability was quantified as the summed orthogonal distances from the forward and backward FSR boundaries, with larger distances indicating less stability.

## Statistical tests

Changes in stability at initial contact (IC), opposite end-of-contact (OEC), and opposite initial contact (OIC) bilaterally were investigated using repeated measures analysis of variance (1 x 4 ANOVA; ɑ = 0.05, two tailed). Normality of the parametric statistical data was assessed by examining Q-Q plots and performing Shapiro-Wilk tests. The Greenhouse-Geisser adjustment was applied to the degrees of freedom in cases of sphericity violations. Statistical analyses were performed using Jamovi software (version 2.3.28.0, Sydney, Australia).

In addition to the group-level analysis, a simple single-subject analysis was conducted to visualize and assess individual participant behaviors. Scatter plots were created to show the change in stability values from the no weight (C1) to weight conditions (C2-C4) for each gait event (IC, OEC, OIC). In these plots, individual participants are represented as dots. The group mean and 95% confidence intervals (CIs) are indicated by horizontal lines. Positive values indicate greater stability compared to the no weight condition, while negative values suggest less stability. This visualization allows for a detailed examination of how each participant's stability changed across conditions, providing insights into individual variances and revealing patterns within the data.

## Results

The results indicated that none of the variables violated the normality assumption. Specifically, the Shapiro-Wilk test yielded *p* values above the alpha threshold of 0.05 for all variables, indicating normal distribution. Additionally, visual inspection of the Q-Q plots revealed that the data points closely adhered to the theoretical quantile line, further supporting the normality of the data.

Despite varying conditions involving different arm weight placements, there were no statistically significant differences observed on stability across the conditions ($p > 0.05$; Fig 1; Table 1). The group of participants consistently demonstrated the ability to control the stability of their COM state across the different weighted conditions.

The single-subject scatter plots revealed that different participants responded differently to the same weighted conditions, with some showing increased stability, others decreased, and still others having changes that fell within the 95% CIs (Figs 2 and 3). This variability highlights the individualized nature of participant responses to the weighted conditions, suggesting a complex interplay of personal factors and condition-specific influences on stability.

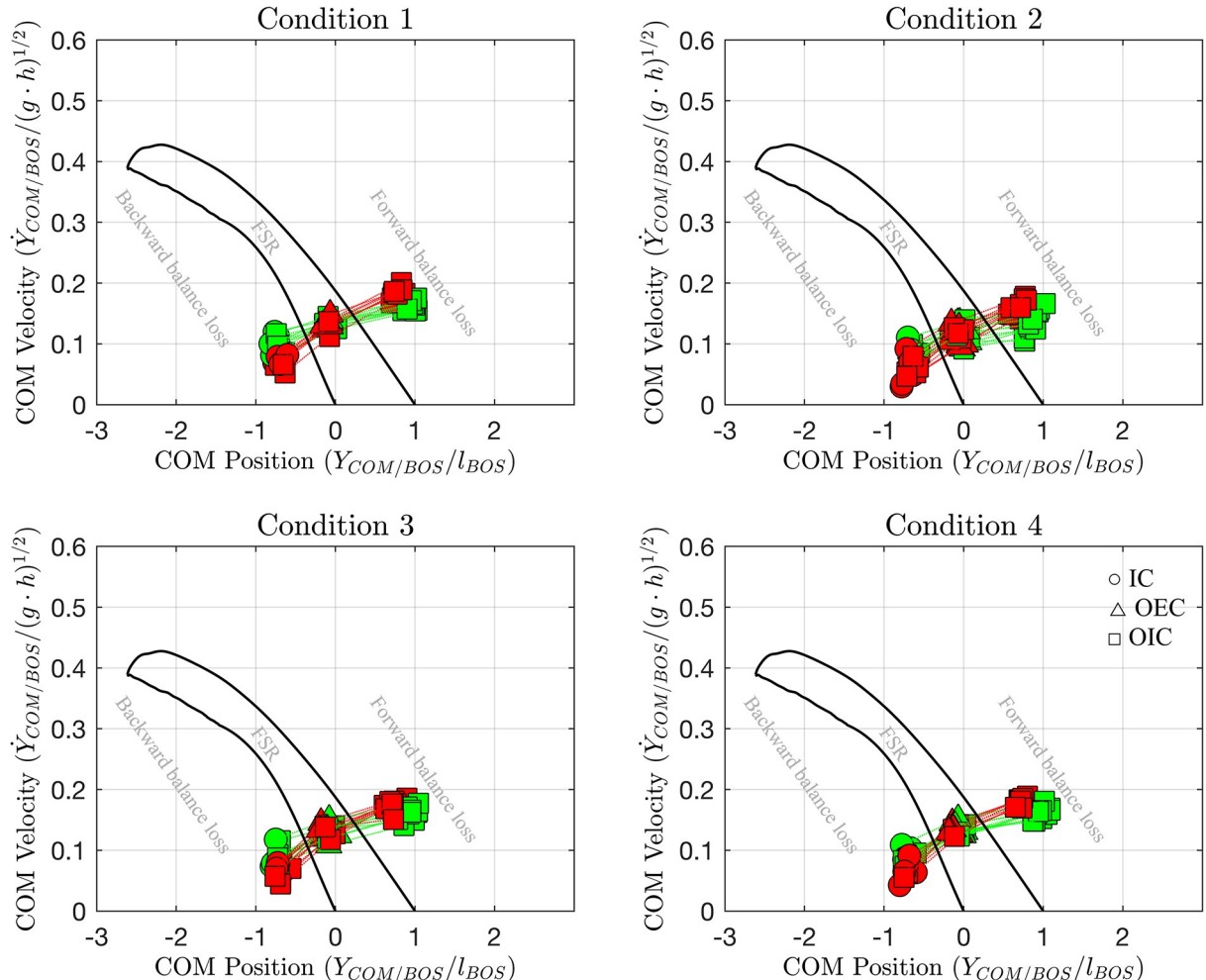

**Fig 1. Dynamic stability based on the Feasible Stability Region (FSR).** Example data from a participant (green: non-hemiparetic side; red: hemiparetic side). Abbreviations: IC = initial contact; OEC = opposite end-of-contact; OIC = opposite initial contact.

## Discussion

The purpose of this study was to investigate the influence of arm weights on dynamic stability in individuals with hemiparetic stroke. The observed findings demonstrate that neither the addition of arm weights nor changes in their placement led to significant differences in

**Table 1. Average stability values (mean ± standard deviation) and statistical results.**

| Stability Measures (unitless) | C1 | C2 | C3 | C4 | $F_{3,24}$ | $\eta^2_p$ | *p* value |
|---|---|---|---|---|---|---|---|
| **HP—IC** | -0.30 ± 0.14 | -0.31 ± 0.11 | -0.30 ± 0.13 | -0.30 ± 0.11 | 0.41 | 0.048 | 0.590 |
| **HP—OEC** | 0.07 ± 0.06 | 0.08 ± 0.06 | 0.09 ± 0.06 | 0.08 ± 0.07 | 1.57 | 0.164 | 0.222 |
| **HP—OIC** | 0.47 ± 0.20 | 0.49 ± 0.20 | 0.52 ± 0.20 | 0.50 ± 0.22 | 2.05 | 0.204 | 0.134 |
| **NHP—IC** | -0.33 ± 0.08 | -0.34 ± 0.08 | -0.34 ± 0.08 | -0.34 ± 0.07 | 0.08 | 0.010 | 0.969 |
| **NHP—OEC** | 0.06 ± 0.06 | 0.07 ± 0.08 | 0.07 ± 0.07 | 0.07 ± 0.07 | 0.90 | 0.102 | 0.454 |
| **NHP—OIC** | 0.53 ± 0.20 | 0.53 ± 0.20 | 0.56 ± 0.20 | 0.56 ± 0.20 | 2.39 | 0.230 | 0.093 |

Abbreviations: C1 = no weight; C2 = non-hemiparetic side arm weight (0.45 kg); C3 = hemiparetic side arm weight; C4 = bilateral arm weights; HP = Hemiparetic side; NHP = Non-hemiparetic side; IC = initial contact; OEC = opposite end-of-contact; OIC = opposite initial contact.

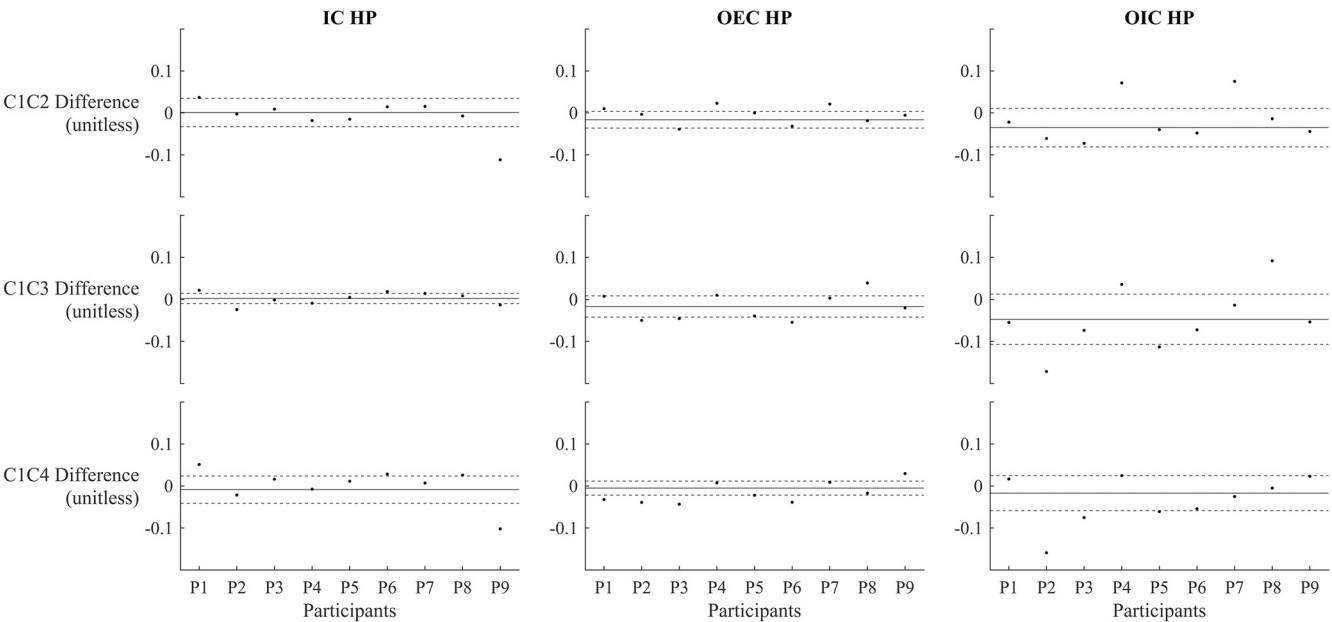

**Fig 2. Stability differences between condition 1 and other conditions across participants on the hemiparetic (HP) side.** The x-axis represents individual participants, and the y-axis shows the absolute stability difference from Condition 1 (unitless). Solid black line: represents the mean difference from Condition 1 across all participants. Dashed lines: indicate the 95% confidence intervals for the mean. Positive values reflect greater stability compared to Condition 1, while negative values suggest reduced stability compared to Condition 1. Abbreviations: IC = initial contact; OEC = opposite end-of-contact; OIC = opposite initial contact.

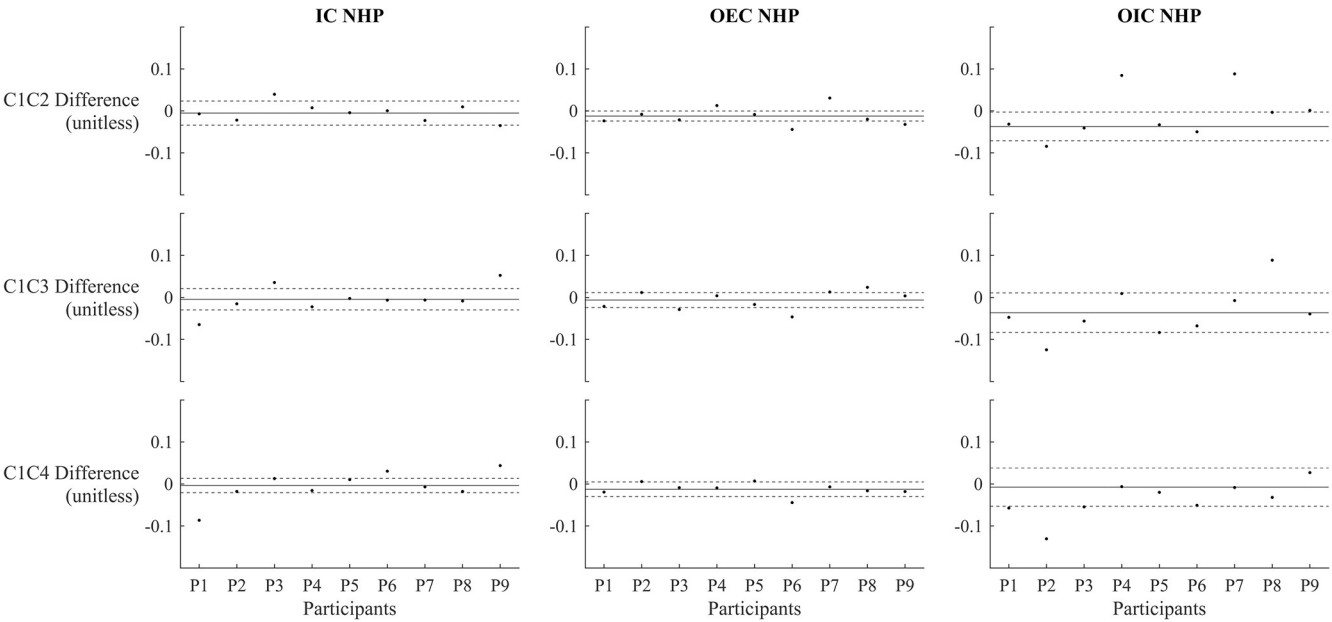

**Fig 3. Stability differences between condition 1 and other conditions across participants on the non-hemiparetic side (NHP) side.** The x-axis represents individual participants, and the y-axis shows the absolute stability difference from Condition 1 (unitless). Solid black line: represents the mean difference from Condition 1 across all participants. Dashed lines: indicate the 95% confidence intervals for the mean. Positive values reflect greater stability compared to Condition 1, while negative values suggest reduced stability compared to Condition 1. Abbreviations: IC = initial contact; OEC = opposite end-of-contact; OIC = opposite initial contact.

stability measures during various gait events. However, when examining individual responses through single-subject scatter plot analyses, the data revealed variability among participants. While overall group analysis did not show significant changes, individual plots (Figs 2 and 3) indicated that some participants experienced increases in stability, whereas others experienced decreases or no change. These results suggest the presence of responders and non-responders to the intervention, highlighting the importance of personalized approaches in rehabilitation planning. This individual variability underscores the complexity of neurorehabilitation, where different stroke survivors may require tailored interventions based on their unique responses to treatment.

The absence of significant differences in stability across the different arm weight conditions suggests that individuals with hemiparetic stroke are capable of maintaining dynamic stability despite the perturbation of adding arm weight/s. This finding aligns with previous studies that have demonstrated the adaptability of stroke survivors during walking [27, 28]. Specifically, our study supports the idea that stroke survivors can modify their walking patterns and strategies to maintain balance and stability, even when external loads are applied. One possible explanation for this ability is the enhanced proprioceptive feedback provided by the arm weights [16]. Previous research has shown that proprioceptive training can significantly improve balance and gait performance in individuals with stroke [29, 30]. The arm weights in this study may have contributed to improved sensory input, allowing participants to adjust their COM more effectively relative to the BOS, thereby maintaining stability.

It is also possible that the proximity of the weights to the COM contributed to the limited effect on dynamic balance. However, because participants were swinging their arms during walking, the weights were not fixed in one location near the COM. Instead, the movement of the arms caused the position of the weights to vary throughout the gait cycle, likely dispersing their influence on stability. This dynamic motion may have lessened the overall impact of the weights on balance. Future studies could explore conditions where arm movement is restricted or weights are placed in different locations to better assess their influence on dynamic stability.

Given the findings of this study, incorporating lighter arm weights into rehabilitation protocols could be a feasible and beneficial strategy. The use of arm weights could potentially enhance proprioceptive feedback without compromising stability, thus supporting the motor re-education and functional training of stroke survivors. This approach could be particularly useful in gait training programs, where the goal is to improve walking performance and reduce fall risk. Additionally, our results suggest that the placement of arm weights (e.g., non-hemiparetic side, hemiparetic side, or both sides) does not significantly alter stability outcomes at the group level. However, single-subject scatter plots indicate variations in response, underscoring the importance of identifying responders and non-responders before universally applying the arm weight intervention. This flexibility in weight placement allows for interventions tailored to individual needs and preferences, potentially increasing adherence to rehabilitation exercises.

Despite these insights, several limitations warrant consideration. The small sample size (n = 9) and high variability among participants may constrain the generalizability of our findings. However, the in-depth analysis of each participant's response across multiple testing scenarios enhances the reliability of the data, suggesting that the trends identified may extend beyond this small group. We also acknowledge that the heterogeneity of the participant group, including differences in rehabilitation history and age, could have contributed to the variability in responses. Participants with varied rehabilitation experiences may have developed different compensatory strategies, and those over 60 years of age could experience age-related declines in balance and proprioception. However, because we used a repeated measures design, where each participant served as their own control, the individual differences were minimized in

their effect on overall outcomes. Still, this variability may have influenced individual responses across conditions.

Additionally, potential errors due to soft tissue artifacts and marker placement may have contributed to variability in the results. Although markers were placed directly on the skin following established protocols, such errors are common in 3D gait analysis. Advanced techniques, such as markerless motion capture, should be considered to minimize these issues in future studies. Furthermore, while our study was limited by its small sample size, further research should investigate whether factors such as age, rehabilitation history, and baseline motor function predict individual responses to arm weight interventions. Identifying these predictors could enable clinicians to tailor rehabilitation programs and enhance the effectiveness of interventions for stroke survivors.

The study also focused on a specific weight (0.45 kg) and did not examine the effects of different weight magnitudes. Investigating the impact of varying weights could help determine the optimal load for enhancing stability without overburdening the participants. Finally, the stability measure in this study was limited to the anteroposterior direction. Stability in the mediolateral direction is equally important, especially for stroke survivors who often experience balance issues in multiple planes of movement. A more comprehensive evaluation could include tasks such as changing walking directions, sloped walking, or stair negotiation to assess the broader impacts of arm weights on dynamic stability.

In conclusion, our findings indicate that the use of 0.45 kg arm weights did not significantly affect the dynamic stability of individuals with hemiparetic stroke at the group level during over-ground walking. However, individual analyses revealed variability in responses, with some participants maintaining or improving stability and others experiencing decreases. This underscores the importance of personalized rehabilitation strategies, as group-level analyses may mask meaningful individual differences. These findings suggest that arm weights could be a viable addition to rehabilitation programs, potentially enhancing proprioceptive feedback and supporting motor recovery without compromising stability. This stability maintenance, coupled with improved walking performance in some individuals, highlights the potential utility of arm weights as an intervention strategy [16]. Further research is warranted to explore the broader applications, optimal parameters, and specific participant profiles for using arm weights in stroke rehabilitation.

## Supporting information

**S1 Data. Individual stability values for different gait events under various conditions.** Each column represents stability measurements at initial contact (IC), opposite end of contact (OEC), and opposite initial contact (OIC) across four conditions (C1-C4) on the hemiparetic (HP) and non-hemiparetic (NHP) sides.
(XLSX)

## Acknowledgments

We express our appreciation to Dr. Feng Yang (Georgia State U.) for sharing data and advice needed to use the FSR method.

## Author Contributions

**Conceptualization:** Hyung Suk Yang, Lee T. Atkins, C. Roger James.

**Formal analysis:** Hyung Suk Yang, C. Roger James.

**Investigation:** Hyung Suk Yang, Lee T. Atkins, C. Roger James.

**Methodology:** Hyung Suk Yang, Lee T. Atkins, C. Roger James.

**Project administration:** Hyung Suk Yang, Lee T. Atkins.

**Supervision:** C. Roger James.

**Visualization:** Hyung Suk Yang.

**Writing – original draft:** Hyung Suk Yang, C. Roger James.

**Writing – review & editing:** Hyung Suk Yang, Lee T. Atkins, C. Roger James.

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
