## [Decision Letter · Decision Letter 0]

21 Oct 2024

PONE-D-24-37984Arm weight effects on dynamic walking stability in individuals with hemiparetic strokePLOS ONE

Dear Dr. Yang,

Thank you for submitting your manuscript to PLOS ONE. After careful consideration, we feel that it has merit but does not fully meet PLOS ONE’s publication criteria as it currently stands. Therefore, we invite you to submit a revised version of the manuscript that addresses the points raised during the review process.

Both reviewers acknowledge the study's value but raise key concerns. Reviewer #1 requests clarification on the choice of the FSR-based balance evaluation method and the rationale for using a 0.45 kg weight. They also suggest discussing how the weight's position, close to the body's center of mass, may have impacted the results. Reviewer #2 highlights the variability due to the heterogeneous patient group and potential errors in 3D gait analysis. They also note contradictions in the discussion's conclusions and recommend further analysis of factors influencing patients' responses to the weights.

We look forward to receiving your revised manuscript.

Kind regards,

Jyotindra Narayan

Academic Editor

PLOS ONE

Journal Requirements:

1. When submitting your revision, we need you to address these additional requirements. Please ensure that your manuscript meets PLOS ONE's style requirements, including those for file naming. The PLOS ONE style templates can be found at https://journals.plos.org/plosone/s/file?id=wjVg/PLOSOne_formatting_sample_main_body.pdf and https://journals.plos.org/plosone/s/file?id=ba62/PLOSOne_formatting_sample_title_authors_affiliations.pdf 2. Please include captions for your Supporting Information files at the end of your manuscript, and update any in-text citations to match accordingly. Please see our Supporting Information guidelines for more information: http://journals.plos.org/plosone/s/supporting-information.

Reviewers' comments:

Reviewer's Responses to Questions

**Comments to the Author**

1. Is the manuscript technically sound, and do the data support the conclusions?

Reviewer #1: Partly

Reviewer #2: Partly

2. Has the statistical analysis been performed appropriately and rigorously? 

Reviewer #1: Yes

Reviewer #2: Yes

3. Have the authors made all data underlying the findings in their manuscript fully available?

Reviewer #1: Yes

Reviewer #2: Yes

4. Is the manuscript presented in an intelligible fashion and written in standard English?

Reviewer #1: Yes

Reviewer #2: Yes

5. Review Comments to the Author

Reviewer #1: This study examined the effects of arm weights on dynamic stability during overground walkingin individuals with hemiparetic stroke. The results indicated that arm weights could be a viable addition to rehabilitation programs, potentially enhancing proprioceptive feedback and supporting motor recovery without compromising stability. More explanation of the analysis method is needed before publication.

1) Why did the authors use the FSR-based balance evaluation method among the many dynamic stabiilty measures? For example, MOS could be used (or is simpler), but it is necessary to explain the reasons why this method (FSR) was used, including the advantages of FSR for this study.

2) Please provide any rationale for the mass of the weight (0.45 kg).

3) The reason why the light weight did not have much effect on the balance this time may be due to the position of the weight. Participants carried a weight by his/her hand, but the position of the weight is close to whole body COM, so it may not have much effect on dynamic balance. If the position and length of the weight were different, the dynamic balance would be more affected. This point should be included in the discussion.

Reviewer #2: I have few concerns connected with this paper:

- Patients: the group is very heterogeneous (the authors correctly listed it as one of the limitations). Two factors of this heterogeneity could influence the results of this study: type and duration of the rehabilitation after stroke received of the patients, and the fact that some of them are over 60 years of age – the natural process of aging starts around this age. This can be the reason of the variability of the effect of weight addition among subjects.

- 3D gait analysis is prone to errors due to the marker placement errors and soft tissues artefacts. From the described methodology I presume that some of the markers were placed on the shirts and shorts adding the errors connected with their movement relative to the patients’ body. This fact can be an additional source of the variability of the obtained results.

- In the Discussion section the authors presented a contradictive (to me) conclusions: that there was no effects of the addition of the weights, and that the analysis of the effect in the individual patients showed in some of them their influence. In my opinion these conclusions should be reformulated.

- I wonder if some analysis could be performed to reveal which factors influence the patients’ response to the weights.

6. PLOS authors have the option to publish the peer review history of their article (what does this mean?). If published, this will include your full peer review and any attached files.

Reviewer #1: No

Reviewer #2: No

---

## [Author Response · Author response to Decision Letter 0]

23 Oct 2024

Please see the Response to Reviewers file.

---

## [Decision Letter · Decision Letter 1]

12 Nov 2024

Arm weight effects on dynamic walking stability in individuals with hemiparetic stroke

PONE-D-24-37984R1

Dear Dr. Yang,

We’re pleased to inform you that your manuscript has been judged scientifically suitable for publication and will be formally accepted for publication once it meets all outstanding technical requirements.

Kind regards,

Jyotindra Narayan

Academic Editor

PLOS ONE

Additional Editor Comments (optional):

The reviewers have recommended the manuscript for the publication. Congratulations to the authors for the significant contribution. 

Reviewers' comments:

Reviewer's Responses to Questions

**Comments to the Author**

1. If the authors have adequately addressed your comments raised in a previous round of review and you feel that this manuscript is now acceptable for publication, you may indicate that here to bypass the “Comments to the Author” section, enter your conflict of interest statement in the “Confidential to Editor” section, and submit your "Accept" recommendation.

Reviewer #1: All comments have been addressed

Reviewer #2: All comments have been addressed

2. Is the manuscript technically sound, and do the data support the conclusions?

Reviewer #1: Yes

Reviewer #2: Partly

3. Has the statistical analysis been performed appropriately and rigorously? 

Reviewer #1: Yes

Reviewer #2: Yes

4. Have the authors made all data underlying the findings in their manuscript fully available?

Reviewer #1: Yes

Reviewer #2: Yes

5. Is the manuscript presented in an intelligible fashion and written in standard English?

Reviewer #1: Yes

Reviewer #2: Yes

6. Review Comments to the Author

Reviewer #1: Thanks for the revision. The authors responded adequately to my comments and revised the manuscript accordingly.

Reviewer #2: The authors in very nice and detailed way responded to all concerns raised in the previous review. All the introduced changes during the review process are marked in the paper's body, so they are easy to indentify. Still, the study is rather a preliminary one and (low number of subjects, the same weight despite the subjects' variability).

7. PLOS authors have the option to publish the peer review history of their article (what does this mean?). If published, this will include your full peer review and any attached files.

Reviewer #1: No

Reviewer #2: No

---

## [Editor Report · Acceptance letter]

14 Nov 2024

PONE-D-24-37984R1 

PLOS ONE

Dear Dr. Yang, 

I'm pleased to inform you that your manuscript has been deemed suitable for publication in PLOS ONE. Congratulations! Your manuscript is now being handed over to our production team.

Kind regards, 

on behalf of

Dr. Jyotindra Narayan 

Academic Editor

PLOS ONE